# Wire Electrical Discharge Machining of AISI304 and AISI316 Alloys: A Comparative Assessment of Machining Responses, Empirical Modeling and Multi-Objective Optimization

**Mona A. Aboueleaz [1], Noha Naeim [2], Islam H. Abdelgaliel [3,4], Mohamed F. Aly [3,*] and Ahmed Elkaseer [2,5,6]**

1  Production Engineering and Mechanical Design Department, Faulty of Engineering, Mansoura University, Mansoura 35516, Egypt; maboueleaz@mans.edu.eg
2  Department of Production Engineering and Mechanical Design, Faculty of Engineering, Port Said University, Port Fuad 42526, Egypt; noha.fouaad@eng.psu.edu.eg (N.N.); ahmed.elkaseer@kit.edu (A.E.)
3  Department of Mechanical Engineering, School of Sciences and Engineering, The American University in Cairo, AUC Avenue, New Cairo 11835, Egypt; ihamdys@aucegypt.edu
4  Department of Mechanical Engineering, Faculty of Engineering, Fayoum University, Fayoum 63514, Egypt
5  Department of Mechanical Engineering, Faculty of Engineering, The British University in Egypt (BUE), El-Sherouk City 11837, Egypt
6  Institute for Automation and Applied Informatics, Karlsruhe Institute of Technology, 76344 Eggenstein-Leopoldshafen, Germany
*  Correspondence: mfawzyaly@aucegypt.edu

**Abstract:** This research investigates the multi-response of both material removal rate (MRR) and surface roughness (Ra) for the wire electrical discharge machining (WEDM) of two stainless steel alloys: AISI 304 and AISI 316. Experimental results are utilized to compare the machining responses obtained for AISI 316 with those obtained for AISI 304, as previously reported in the literature. The experimental work is conducted through a full factorial experimental design of five running parameters with different levels: applied voltage, transverse feed, pulse-on/pulse-off times and current intensity. The machined workpieces are analyzed using an image processing technique in order to evaluate the size of cut slots to allow the calculation of the MRR. Followed by the characterization of the surface roughness along the side walls of the slots. Different mathematical regression techniques were developed to represent the multi-response of both materials using the MATLAB regression toolbox. It was found that WEDM process parameters have a fuzzy influence on the responses of both material models. This allowed for multi-objective optimization of the regression models using four different techniques: multi-objective genetic algorithm (MOGA), multi-objective pareto search algorithm (MOPSA), weighted value grey wolf optimizer (WVGWO) and osprey optimization algorithm (OOA). The optimization results reveal that the optimal WEDM parameters of each response are inconsistent to the others. Hence, the optimal results are considered a compromise between the best results of different responses. Noteworthily, the multi-objective pareto search algorithm outperformed the other candidates. Eventually, the optimal results of both materials share the high voltage, high transverse feed rate and low pulse-off time parameters; however, AISI 304 requires low pulse-on time and current intensity levels while AISI 316 optimal results entail higher pulse-on time and current levels.

**Keywords:** WEDM; stainless steel 304; stainless steel 316; surface integrity; MRR; productivity; multi-objective optimization

## 1. Introduction

The challenging issue of manufacturing functional products with high accuracy and superior mechanical and thermal properties encourage several researchers to develop new material processing techniques or improve the existing ones [1,2]. In return, this leads to the necessity of the optimization of process parameters in order to obtain the highest

quality and productivity [3]. Each material cutting process has its own effect on the material structure and properties. In addition, the capability of the cutting methods does not have to include the processing of all material types as some materials require high cutting force or power [4]. Wire electrical discharge machining (WEDM) is one of the key thermo-electric processes of cutting materials, irrespective of their high hardness and strength properties [5,6], such as high-carbon steels [7,8], stainless steel 316 (AISI 316) [9], etc. In WEDM, the cutting process is carried out by means of eroding the workpiece (as a cathode) with a travelling thin wire tensioned between two guide rollers (as an anode) by initiating a series of sparks between the electrodes in the presence of a dielectric medium [10,11]. One of the advantages of WEDM is to process difficult-to-cut materials, irrespective of their hardness, with acceptable surface properties and accurate dimensions, which promotes WEDM as a prime cutting method to produce punch dies and molds of high-strength materials [1,8,10].

There are a wide range of WEDM controlling parameters, such as servo voltage, pulse-on/pulse-off times, transverse feed rate, peak current, machined material thermo-physical properties, wire tension, wire material, wire diameter, and dielectric fluid type and pressure [12–18]. The most challenging problem in WEDM is that most of the mentioned controlling parameters have a fuzzy influence on the desired output, such as surface roughness (Ra) and material removal rate (*MRR*). In return, the process is considered a stochastic problem that requires sufficient mathematical representation and optimization [19–22]. However, using a mathematical model and MATLAB simulated annealing (SA) algorithm, it is claimed that feed rate, peak current and pulse-one time are the most influential parameters of the WEDM of EN31 high-strength steel alloy [23].

Sharma et al. [14] used Taguchi $L_9$ and ANOVA to analyze the *MRR*, gap current and machining time responses of D2 tool steel. It was found that pulse-off time was the most influential and significant parameter for all responses, followed by pulse-on time, while the peak current and wire tension influence were negligible. Using the amalgamation of response surface modeling (RSM), genetic algorithm (GA) and ANOVA, Sharma et al. [24] obtained the optimal WEDM parameters of the minimum overcut (9.9922 µm) of machined high-strength low-alloy (HSLA) steel using brass wire as an electrode at pulse-on/pulse-off times of 117 and 50 µs, respectively; gap voltage of 49 V; current of 180 A; and wire tension of 6 g. Meanwhile, using Taguchi $L_9$, analysis of variance (ANOVA) and signal-to-noise (S/N) ratio, the optimal WEDM process parameters of machining AISI 1045 alloy were obtained. The most influential parameter is current, with a *p*-value of 0.026. Moreover, the best *MRR* is 0.7112 mm$^3$/min, which was obtained at current = 16 A, voltage = 50 V and pulse-on time = 100 µs. The authors recommended further experimental-based multi-objective optimization that includes more responses and variable parameters [25]. Involving eight parameters and using a larger Taguchi array ($L_{18}$ $2^1 \times 3^7$) followed by the ANOVA and S/N ratio, the optimal cutting conditions of Skd 61 alloy steel were obtained separately for the *MRR* and Ra—64.79 mm$^2$/min and 1.279 µm, respectively—as two single objectives, with a maximum relative error to the experimental data that did not exceed 9.8%. In addition, the study stated that pulse-on time is the most influential factor on both *MRR* and Ra models [26]. Additionally, Huang and Liao [27] confirmed that pulse-on time has the main influence on both *MRR* and Ra using grey relational analysis (GRA) and S/N ratio. In a recent study using similar methods, Kumar et al. [28] reduced the number of process parameters to four. In addition, the optimal parameters of the best *MRR* of D2 steel were a wire speed of 21 m/min, flushing pressure of 159 kgs/m$^2$, voltage of 81 V and current of 61 A, while the best Ra was obtained at 31 m/min, 121 kgs/m$^2$, 82 V and 80 A, with the same aforementioned order.

Furthermore, an optimization study using GRA on the WEDM machining of tungsten carbide material found that the optimal surface roughness (0.3435 µm) can be obtained at current = 2 A, voltage = 5 V and pulse-off time = 8 µs [29]. Chen et al. [30] used a combined multi-objective algorithm based on support vector machine and particle swarm optimization (SVR-PSO) to obtain the minimum Ra of 3.6 µm and the maximum *MRR*

of 0.261 mm$^2$/s of 65 vol.% SiCp/Al composite. The optimal WEDM parameters were pulse-on time = 258.2 ns, pulse-off time = 125.3 ns, voltage = 41 V, wire feed = 9.452 mm/s and wire tension = 12.7 N. To wrap up, a review study on the optimization techniques of WEDM used in the literature showed that most researchers are mainly interested in multi-responses such as *MRR*, Ra and tool wear rate (TWR). The authors also stated that the most used optimization technique is Taguchi's methodology followed by GRA and fuzzy logic. Eventually, it was confirmed that current is a dominant factor requiring optimal control in order to obtain optimal multi-response answers [31].

The aim of this study is to widely investigate and optimize WEDM process parameters in order to achieve the trade-off between maximizing the *MRR* and keeping the Ra at the minimum level of two different steel alloys: stainless steels 304 and 316 (AISI 304 and AISI 316). A previously reported experimental investigation of WEDM of AISI 304 is experimentally extended for AISI 316, for which five parameters are investigated and modeled using the MATLAB regression toolbox. A combination of four multi-objective algorithms is used and compared. The algorithms are: (1) multi-objective genetic algorithm (MOGA), (2) multi-objective pareto search algorithm (MOPSA), (3) weighted value grey wolf optimizer (WVGWO) and (4) osprey optimization algorithm.

## 2. Materials and Methods

As formerly mentioned, this is a significant extension of the work reported by Naeim et al. in [10], in which the MRR and obtainable surface roughness in the WEDM of AISI 304 were analyzed. Herein, AISI 316 samples are machined under similar cutting conditions to those reported in [10] with the aim of comparatively assessing the machining responses of both materials under similar conditions. In addition, statistical analyses of the experimental results for both materials are carried out and discussed. The materials' compositions and properties are presented in depth. In addition, the equipment setup of the WEDM process is illustrated. The details of the design of the experiment are also discussed. Finally, the measurement instruments and conditions are provided.

### 2.1. Materials

Stainless steel is the most commonly utilized material for contact surfaces in dairy processing equipment. This metal possesses corrosion resistance, mechanical strength, hardness and ease of manufacture (weldability). AISI types 304 and 316 are the most suitable grades for general process fluid heating, storage and distribution. Because of the presence of molybdenum, type 316 is more expensive but offers superior corrosion resistance. Given that it is primarily required to protect the machine from the atmosphere, water and any spilled liquids, AISI 304 is almost always utilized externally or for the outside vessel jacket. Not only are they known for their resistance to corrosion but also for their clean appearance and overall cleanliness [32].

The experiments were conducted on AISI 304 and AISI 316 specimens with geometry of 120 mm × 30 mm × 3.1 mm. The chemical compositions of both materials are listed in Table 1. The physical and mechanical properties of both materials are listed in Table 2.

**Table 1.** Chemical compositions of used materials (%).

| Grade | Mn | C | S | P | Si | Ni | Cr | Mo | N | V | Fe |
|---|---|---|---|---|---|---|---|---|---|---|---|
| AISI 304 [10] | 2.00 | 0.08 | 0.03 | 0.045 | 0.75 | 8 | 18–20 | - | 0.10 | - | Balance |
| AISI 316 [33] | 1.97 | 0.077 | 0.005 | 0.0004 | 0.49 | 10.18 | 17.13 | 1.853 | - | 0.0615 | Balance |

**Table 2.** Physical and mechanical properties of used materials [34].

| Property | AISI 304 | AISI 316 |
|---|---|---|
| Density (g/cm$^3$) | 8.00 | 8.00 |
| Melting Point (°C) | 1450 | 1400 |
| Modulus of Elasticity (GPa) | 193 | 193 |
| Electrical Resistivity (Ω·m) | $0.72 \times 10^{-6}$ | $0.74 \times 10^{-6}$ |
| Thermal Conductivity (W/m·K at 100 °C) | 16.2 | 16.3 |
| Thermal Expansion ($10^{-6}$/K at 100 °C) | 17.2 | 15.9 |

## 2.2. Equipment Setup

The WEDM machine used is an ONA NX3, which has a positioning resolution of 1 μm. The cutting wire is molybdenum with diameter 0.18 mm. The wire tension is 8 (index). Water and gel were used as dielectrics. For preprocessing, the specimen surface is machined by grinding using a silicon carbide abrasive wheel in order to ensure the specimen's top and bottom surfaces are flat and parallel in all experiment trials (see Figure 1).

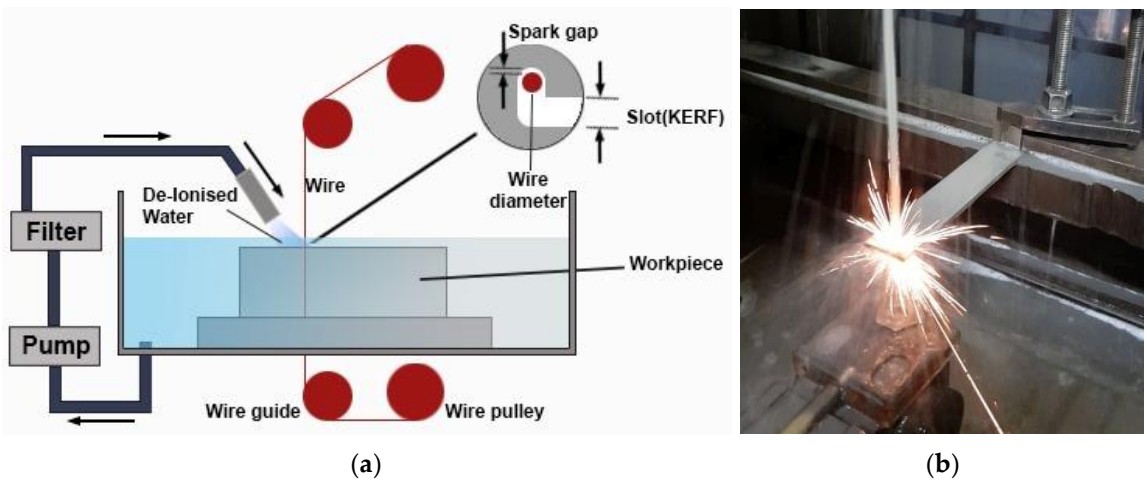

(**a**)          (**b**)

**Figure 1.** Experiment setup of WEDM: (**a**) schematic illustration of WEDM process and (**b**) real experiment on ONA NX3 EDM machine.

## 2.3. Design of Experiment

The selected running conditions for the WEDM, as reported in [10], are voltage (*V*) in V, transverse feed (*f*) in mm/min, pulse-off time ($P_{off}$) in μs, pulse-on time ($P_{on}$) in μs and current intensity (*C*) in A. Each parameter has certain levels, from 2 to 3 levels, as shown in Table 3.

**Table 3.** WEDM selected parameters and their levels.

| Parameter | Levels | | |
|---|---|---|---|
| Voltage (*V*), V | | Low | High |
| Transverse feed (*f*), mm/min | | 80 | 120 |
| Pulse-off time ($P_{off}$), μs | | 6 | 7 |
| Pulse-on time ($P_{on}$), μs | 25 | 30 | 40 |
| Current intensity (*C*), A | 1 | 2 | 4 |

The design of experiment (DOE) used in this research is a full factorial design. This led to conducting 72 experimental runs for both 304 and 316 stainless steels. Each run

was carried out 3 times and the average of trial results was considered. The order of the 72 experiment's running conditions is illustrated in Figure 2.

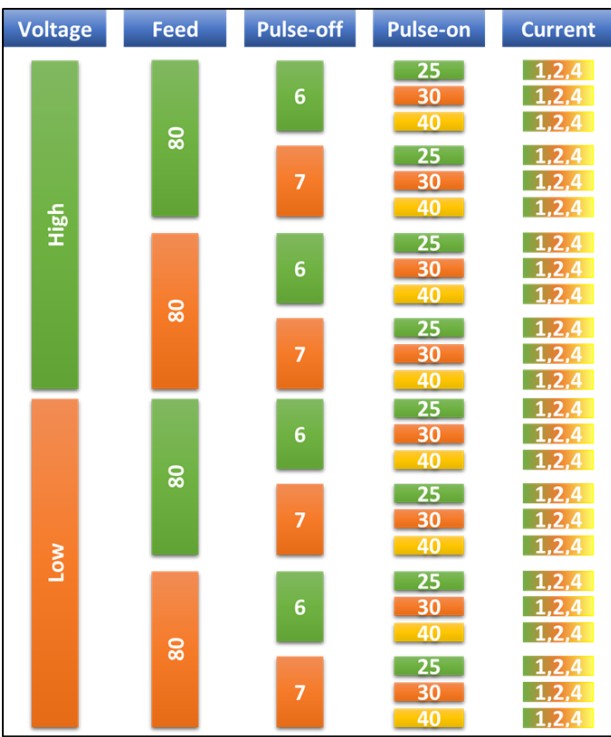

**Figure 2.** Full factorial DOE of experiment.

*2.4. Measurement and Characterization*

Each experimental trial of a certain set of cutting parameters had two wire cuts. The first cut was 15 mm in length and was machined in order to evaluate the cut width. The machined specimens were photographed by a high-resolution camera under appropriate lighting conditions. Following that, the collected images were processed and analyzed using the image processing techniques presented in [10] to detect the edge and precise width of the cut slot along its entire length. In addition, the machining time was calculated alongside the cut width. This allowed the material removal rate (*MRR*) to be calculated. The second cut was a full-width cut of 30 mm that allowed the characterization of the surface quality of the side walls of the cut. A Mitutoyo SJ-210 surface profilometer (Mitutoyo Corp., Kawasaki, Japan) was used to measure the surface roughness (Ra) along the full length of the side walls. To eliminate measurement uncertainty, seven readings were measured for each specimen and the average measurement was considered; further details of the characterization procedures can be found in [10].

## 3. Results and Discussion

### 3.1. Experimental Results

Figures 3 and 4 illustrate the experimental results of the *MRR* and Ra of the AISI 304 alloy. Remarkably, the *MRR* increases proportionally with the increase in the feed rate of 120 mm/min and high voltage. Also, the *MRR* is higher at $P_{off}$ = 6 µs than at 7 µs, as shown in Figure 3c,d. Figure 3c–h show the *MRR* behavior to the change of the $P_{on}$ and C. The *MRR* tends to achieve the best results at high $P_{on}$ and C on the top-right corner of the contours. The *MRR* can reach 7 mm³/min at high voltage, $f$ = 120 mm/min, $P_{off}$ = 6 µs, $P_{on}$ = 40 µs and C = 2.4 A, as shown in Figure 3c. On the contrary, Figure 4a–h show that the Ra is tailored with the best surface quality at low $P_{on}$ and C on the bottom-left corner of the contours. This leads to the best surface quality of Ra = 3.39 µm at high voltage and at the lowest level of the remaining parameters, as shown in Figure 4a.

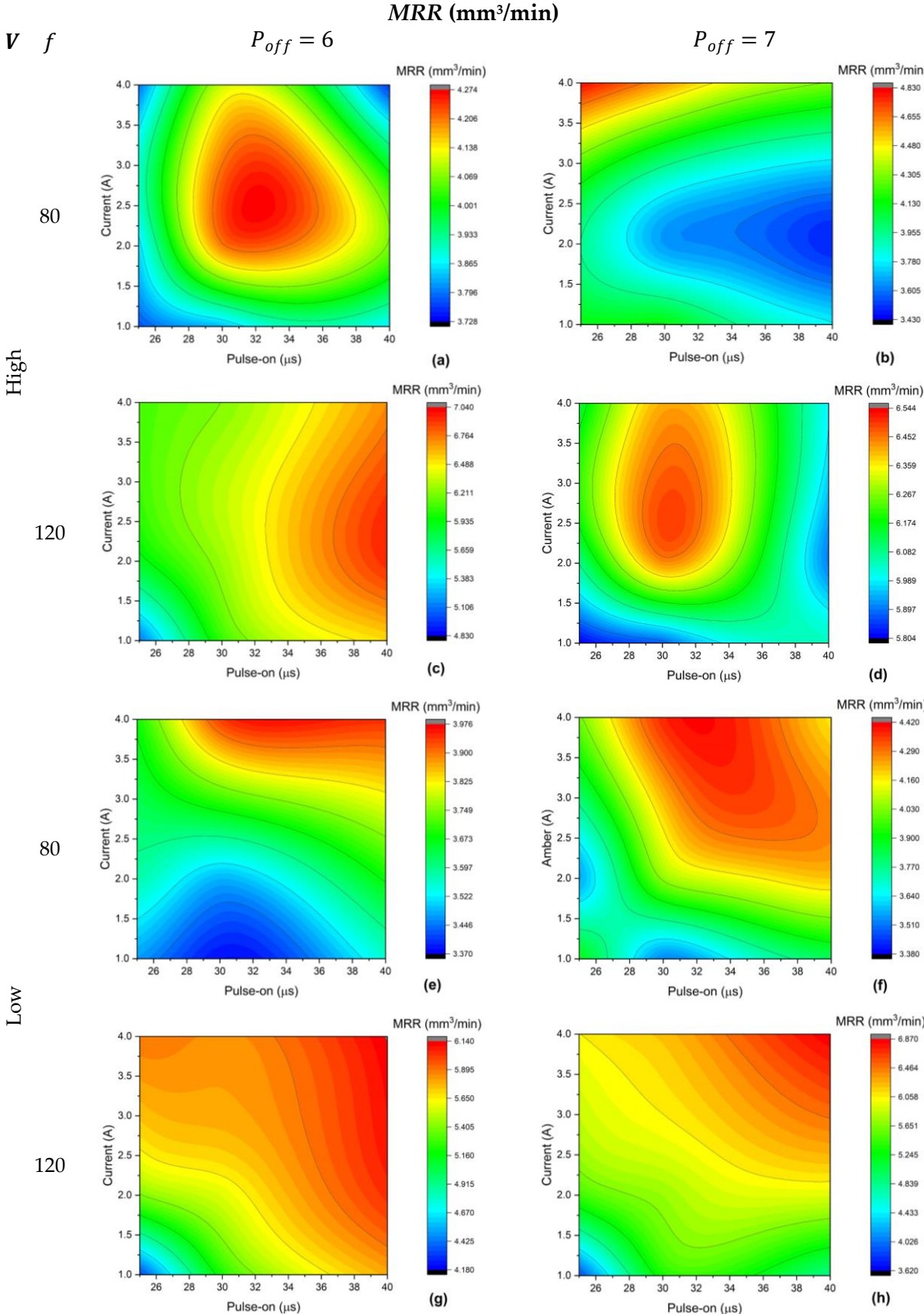

**Figure 3.** Experimental *MRR* results of AISI 304; labels (**a**–**h**) according to the given matrix.

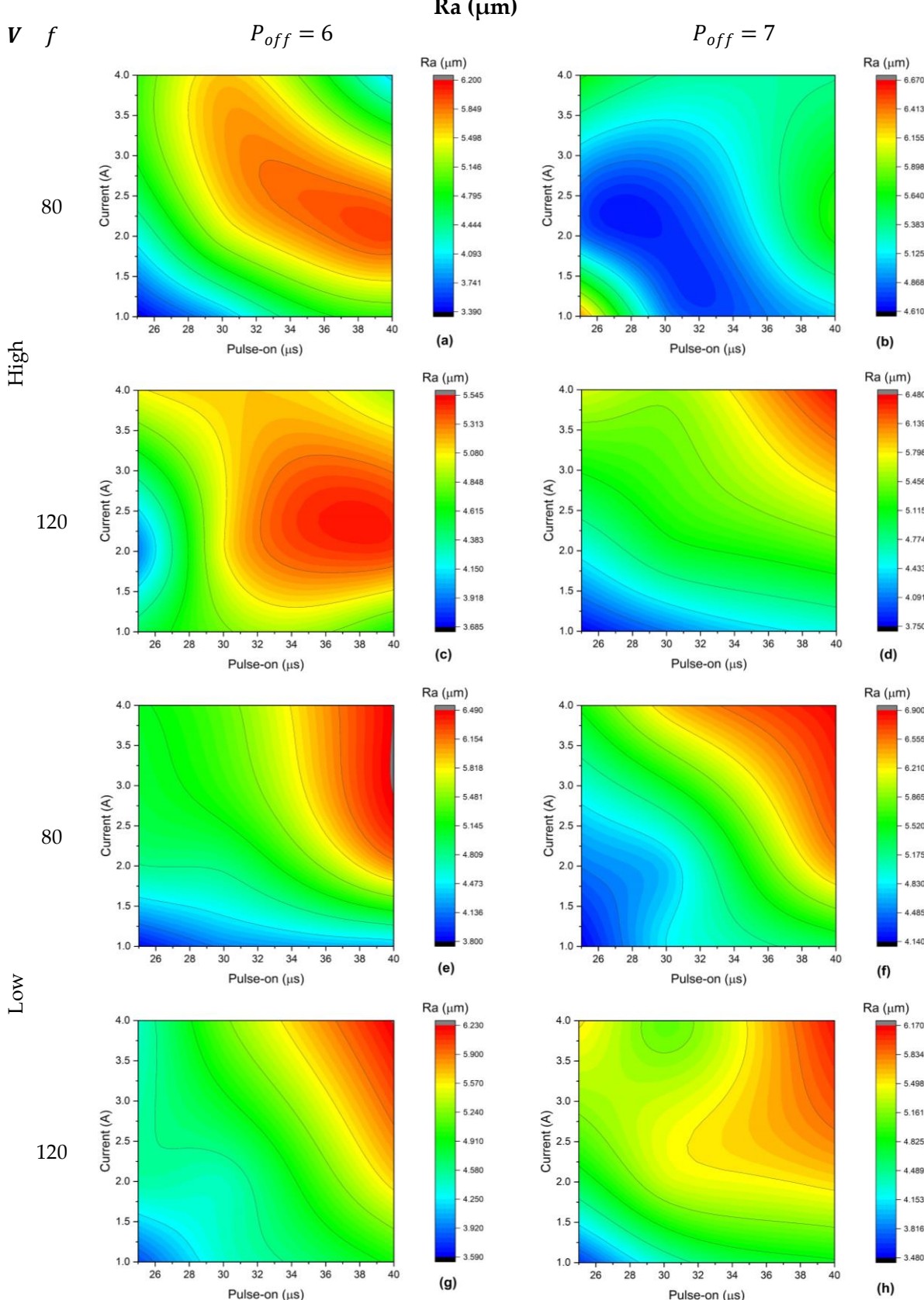

**Figure 4.** Experimental Ra results of AISI 304; labels (**a**–**h**) according to the given matrix.

Figures 5 and 6 explain the parameters' effect on the *MRR* and Ra of the AISI 316 alloy. Again, the *MRR* is high at the upper bounds of voltage, feed rate, pulse-on time and current; conversely, low pulse-off time is recommended for higher productivity of AISI 316, as shown in Figure 5a–d. Hence, the *MRR* entails the best productivity at 6.59 mm³/min, with the corresponding parameters as shown in Figure 5c. Regarding the surface quality of wired AISI 316, the most desired surface quality (3.64 μm) can be obtained at high voltage, low feed rate, low pulse-on and pulse-off times and low current, as shown in Figure 6a.

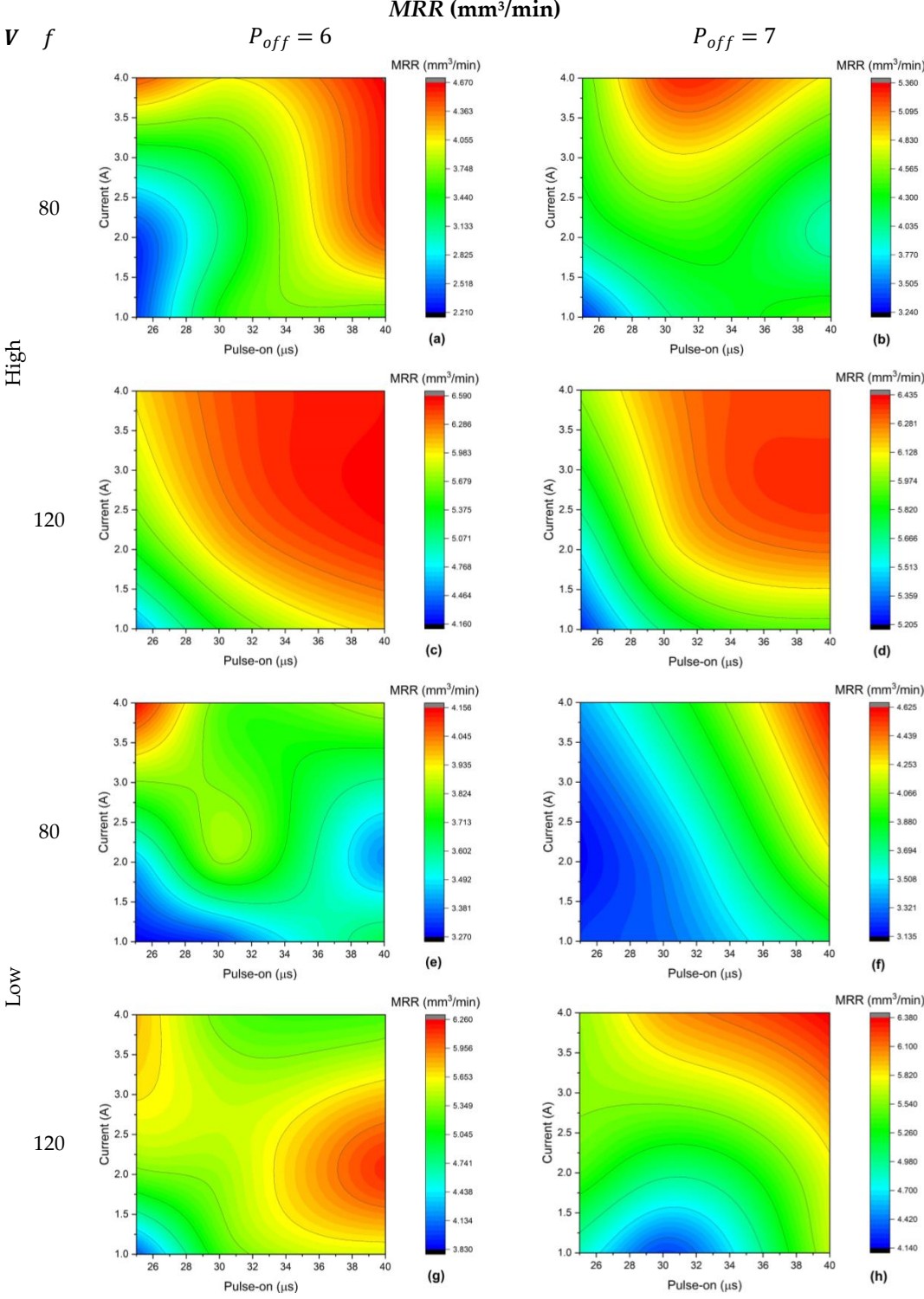

**Figure 5.** Experimental *MRR* results of AISI 316; labels (**a**–**h**) according to the given matrix.

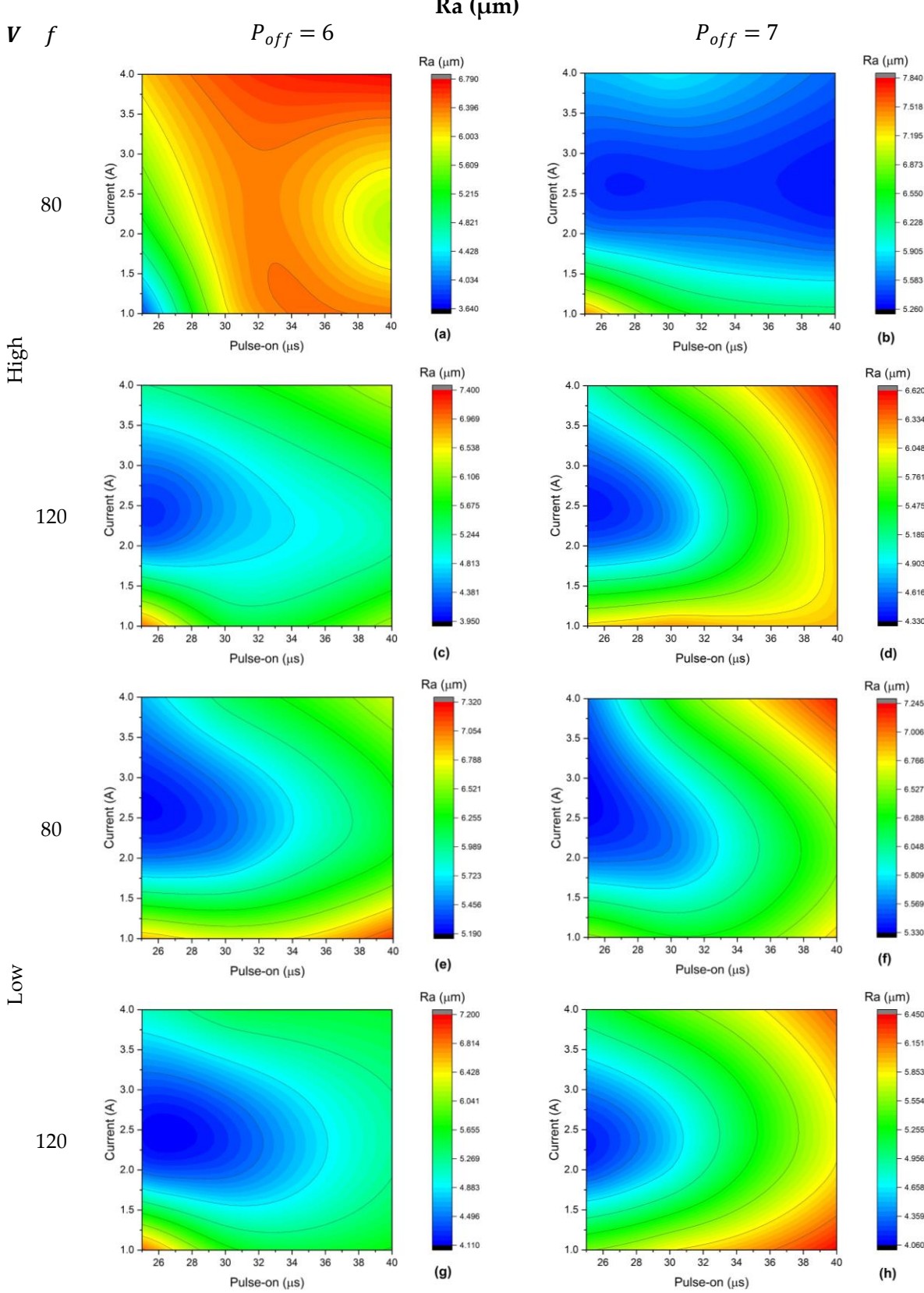

**Figure 6.** Experimental Ra results of AISI 316; labels (**a**–**h**) according to the given matrix.

### 3.2. Mathematical Model Regression

The mathematical models of both used materials are developed using the MATLAB regression learner toolbox. In order to obtain close accurate fitting, the parameters are normalized to the interval $[-1,1]$ using Equation (1). Hence, the subscript $n$ stands for the normalized parameter (see Table 4).

$$x_n = 2\left(\frac{x - x_{min}}{x_{max} - x_{min}}\right) - 1 \tag{1}$$

**Table 4.** WEDM normalized parameters and their new levels.

| Parameter | Levels | | |
|---|---|---|---|
| Voltage ($V$), V | $-1$ | 1 | |
| Transverse feed ($f$), mm/min | $-1$ | 1 | |
| Pulse-off time ($P_{off}$), μs | $-1$ | 1 | |
| Pulse-on time ($P_{on}$), μs | $-1$ | $-0.3333$ | $-1$ |
| Current intensity ($C$), A | $-1$ | $-0.3333$ | $-1$ |

### 3.2.1. AISI 304 Model

MATLAB quadratic regression is used to develop the material removal rate ($MRR$) model of AISI 304, as shown in Equation (2). The values of the R-squared = 90.1% and R-adjusted = 88.3%. Meanwhile, the model $p$-value = $5.92 \times 10^{-26}$. Also, a comparison between the developed model and the experimental trials is conducted, as shown in Figure 7. It was found that the maximum error between the developed model and the experiment results is 36.47% and the average error is 5.74%. The maximum error appeared in one trial that the model was incapable of fitting with.

$$\begin{aligned} MRR_{304} = {}& 4.9286 + 0.17772\,V_n + 1.0069\,f_n + 0.037764\,P_{off_n} + \\ & 0.14666\,P_{on_n} + 0.28073\,C_n + 0.085168\,V_n f_n - 0.11894\,V_n C_n - \\ & 0.0766\,f_n P_{off_n} + 0.14842\,f_n P_{on_n} + 0.093122\,f_n C_n - 0.099665\,P_{off_n} P_{on_n} \end{aligned} \tag{2}$$

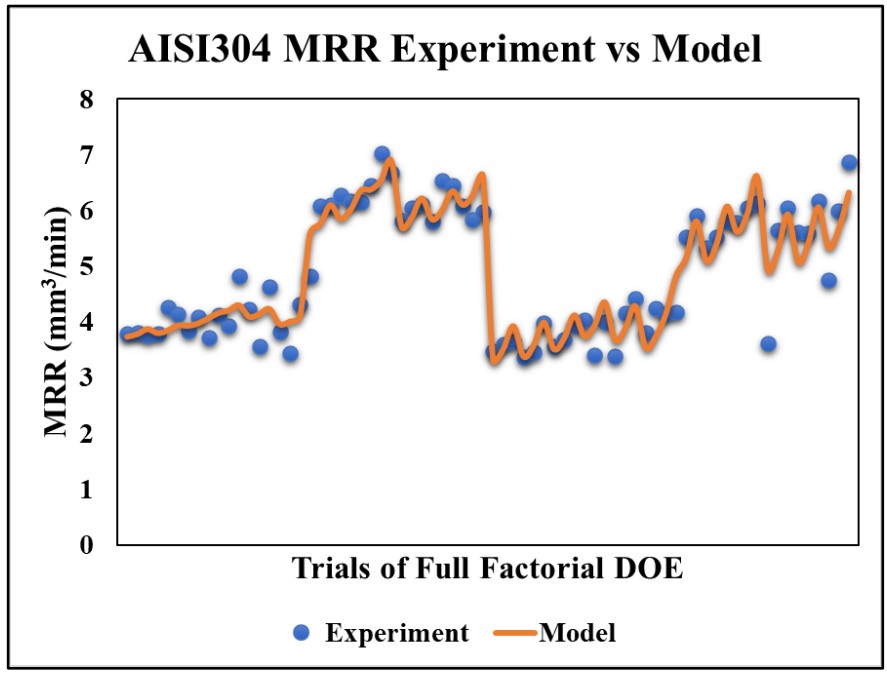

**Figure 7.** AISI 304 *MRR* model vs. experiment results.

For the roughness (Ra) model, the linear stepwise regression method on MATLAB is used (see Equation (3)). The fitting is quite inappropriate, as the R-squared = 60% and R-adjusted = 50%; however, the model $p$-value is $5.64 \times 10^{-6}$, which is a good indicator of significance. The comparison between the regression model and the actual experiment results, shown in Figure 8, revealed that the maximum error is 34.04% while the average error increased to 7.88% due to the low R-squared and R-adjusted values. To wrap up, the model can be incapable of fitting with the exact values of the experimental results; however, the model maintained the proportionality of results in a desired shape.

$$
\begin{aligned}
Ra_{304} = {} & 5.3725 - 0.090425\, V_n - 0.091921\, f_n + 0.17402\, P_{off_n} + \\
& 0.40703\, P_{on_n} + 0.51585\, C_n + 0.044755\, V_n f_n + 0.028745\, V_n P_{off_n} - \\
& 0.1892\, V_n P_{on_n} - 0.13684\, V_n C_n - 0.091468\, f_n P_{off_n} + 0.076865\, P_{off_n} C_n - \\
& 0.055949\, P_{on_n} C_n - 0.33842\, C_n{}^2
\end{aligned}
\tag{3}
$$

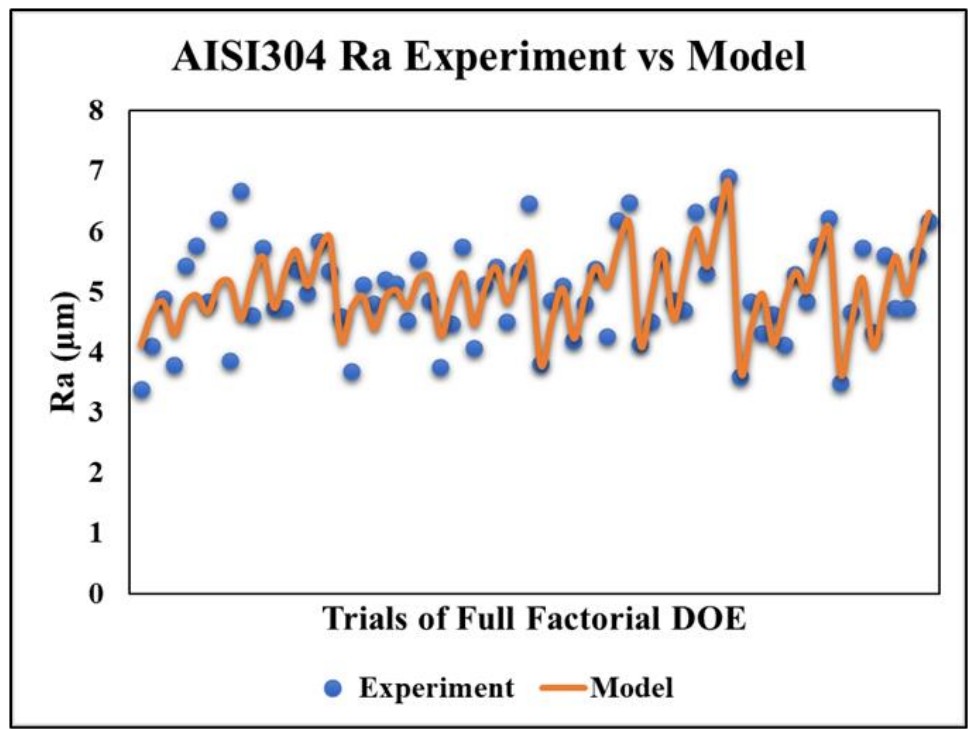

**Figure 8.** AISI 304 Ra model vs. experiment results.

### 3.2.2. AISI 316 Model

The *MRR* model of the second material AISI 316 is developed using the MATLAB linear interaction method, which is considered a first-order linear regression, as in Equation (4). Figure 9 depicts a comparison between the model and the experiment with R-squared = 87.8%, R-adjusted = 84.5% and $p$-value = $3.53 \times 10^{-20}$. The calculated errors are maximum error = 21% and average error = 7.77%. Fruitfully, this promotes the model to be reliable for further calculations.

$$
\begin{aligned}
MRR_{316} = {} & 4.7202 + 0.20988\, V_n + 0.91948\, f_n + 0.10544\, P_{off_n} + \\
& 0.34751\, P_{on_n} + 0.39651\, C_n + 0.049668\, V_n f_n + 0.073788\, V_n P_{off_n} + \\
& 0.072056\, V_n P_{on_n} + 0.072247\, V_n C_n - 0.061812\, f_n P_{off_n} - 0.18745\, P_{on_n} C_n
\end{aligned}
\tag{4}
$$

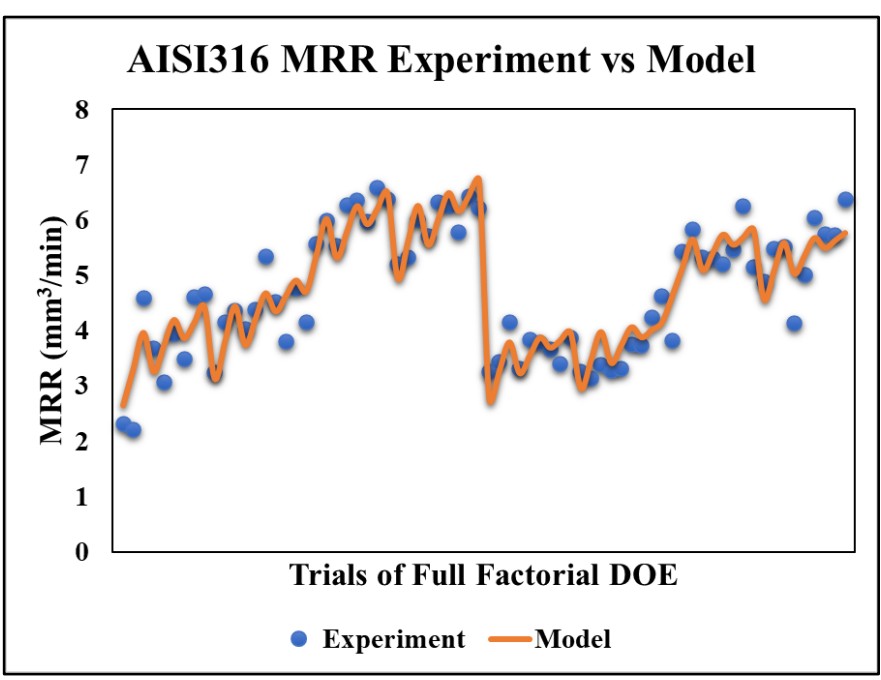

**Figure 9.** AISI 316 *MRR* model vs. experiment results.

Equation (5) is obtained by using the MATLAB linear stepwise regression method. The previously calculated values—R-squared = 61%, R-adjusted = 60%, *p*-value = $2.24 \times 10^{-10}$, maximum error = 21% and average error = 6.44%—are obtained. Again, the Ra model slightly lacks a fit with the experiment; however, the fitting is in good proportion, as shown in Figure 10.

$$Ra_{316} = 5.0791 - 0.031417\, V_n - 0.28337\, f_n + 0.05213\, P_{off_n} + \\ 0.32202\, P_{on_n} - 0.21772\, C_n + 0.13506\, V_n f_n + 0.25158\, P_{on_n} C_n + 0.99187\, C_n{}^2 \tag{5}$$

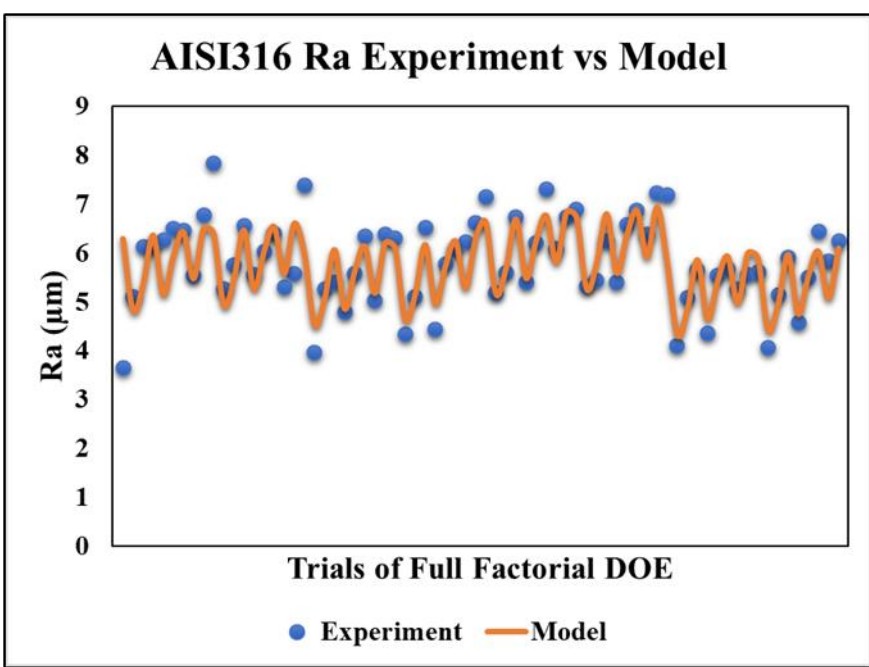

**Figure 10.** AISI 316 Ra model vs. experiment results.

### 3.3. Multi-Objective Optimization Results

In this stage, the developed models in the previous Section 3.2 are promoted to be used as objective functions in the multi-objective optimization model. In order to equalize the optimization results for a better solution, the model results of *MRR* and Ra are normalized from 0 to 1. Hence, the new symbols $MRR_n$ and $Ra_n$ are used. Then, the objective functions minimize $Ra_n$ and maximize $MRR_n$. Almost all optimization algorithms tend to minimize the objective function; although, the minimization of $(1-MRR_n)$ is considered. The lower and upper bounds of the parameters are set to be $-1$ and 1, respectively, as the inputs parameters are normalized. However, the voltage parameter is forced to be equal to $-1$ (if $V_n \leq 0$) for a low value and 1 (if $V_n > 0$) for a high value. In this work, four multi-objective algorithms are used; two of them are built-in MATLAB functions—the multi-objective genetic algorithm (MOGA) [35,36] and multi-objective pareto search algorithm (MOPSA) [37–39]—while the other two are brought from the literature—the weighted value gray wolf optimizer (WVGWO) [40,41] and osprey optimization algorithm (OOA) [42]. The MOGA, MOPSA and WVGWO algorithms showed impressive solutions in recent work by the authors of [43,44]. Meanwhile, the OOA was introduced in 2023, and the author claimed that OOA outperformed many recent and late, well-known algorithms [42]. The optimization models of both materials are quite similar for all algorithms, as shown in Table 5.

**Table 5.** The multi-objective optimization model.

| Model Item | Values |
|---|---|
| Number of Variables | 5 |
| Lower Bounds | $\begin{pmatrix} V_n & f_n & P_{off_n} & P_{on_n} & C_n \\ -1 & -1 & -1 & -1 & -1 \end{pmatrix}$ |
| Upper Bounds | $\begin{pmatrix} V_n & f_n & P_{off_n} & P_{on_n} & C_n \\ 1 & 1 & 1 & 1 & 1 \end{pmatrix}$ |
| Linear Inequality | [ ] |
| Linear Equality | [ ] |
| Initial Starting Point | $\begin{pmatrix} V_n & f_n & P_{off_n} & P_{on_n} & C_n \\ 0 & 0 & 0 & 0 & 0 \end{pmatrix}$ |
| Objective Function 1 | Min $(1-MRR_n)$ |
| Objective Function 2 | Min $(Ra_n)$ |

### 3.3.1. MOGA Model

The two material models are inserted into the MOGA toolbox in MATLAB. The additional options are maximum stall generations, set to 1000, and the maximum generations are 2000. In fact, the MOGA's output is a 2D x–y plot of the two normalized objectives, see Table 5, on both axes considering all data on the plot as a solution. This graph shows the trade-off between the two objectives—that is, the so called "Pareto front". Figure 11 illustrates the Pareto front graph of both AISI 304, Figure 11a, and AISI 316, Figure 11b. The gold dash box surrounds the desirable solution region, which is called the "Feasible Solution Area". One can resize the feasible solution area as required, for example, the box is set to be [0.4, 0.4] in size. The optimal solution is selected depending on the furthest point on a parallel line to a trend line connecting the plot data. There is a remarkable difference between the two materials' optimal solutions. Both optimal solutions are triggered at high voltage. The AISI 304 model is optimal at $f$ = 119.37 mm/min, $P_{off}$ = 6.16 µs, $P_{on}$ = 28.66 µs and $C$ = 1.06 A; in addition, AISI 316 achieved the optimal solution at $f$ = 119.97 mm/min, $P_{off}$ = 6.06 µs, $P_{on}$ = 39.53 µs and $C$ = 3.36 A. At these running conditions, the *MRR*s of AISI 304 and AISI 316 are 5.707 mm³/min and 6 mm³/min, respectively. Also, the surface roughness (Ra) is 4.353 µm and 4.741 µm for the same mentioned order. The *MRR* of AISI 316 is 5% better than that of AISI 304; however, the surface quality is reduced by 9%.

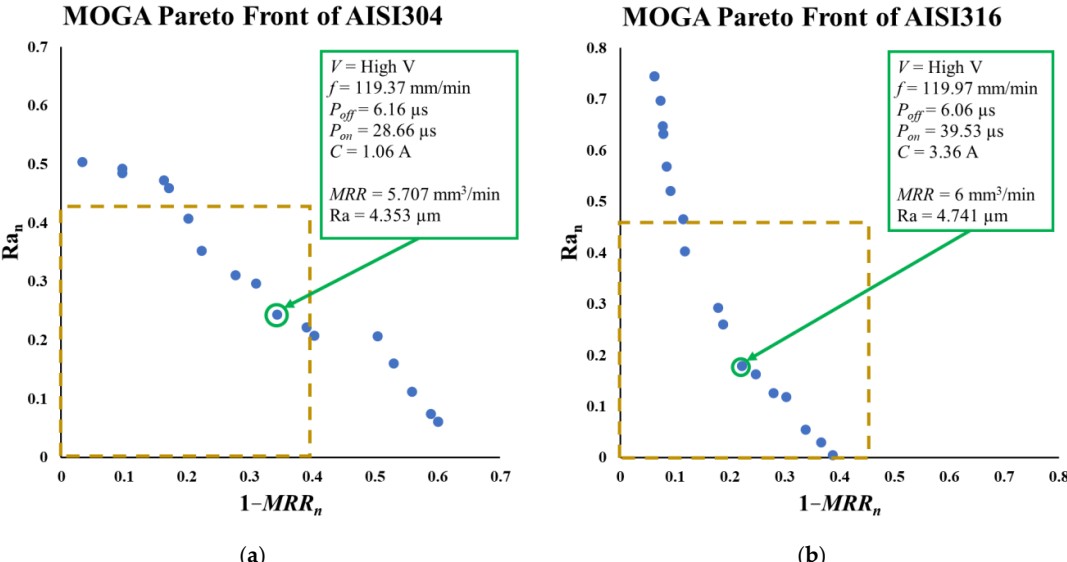

**Figure 11.** Optimal solutions by MOGA Pareto front of both (**a**) AISI 304 and (**b**) AISI 316.

### 3.3.2. MOPSA Model

Similar to the MOGA, the input parameters in Table 5 are the same except for the options used. The maximum iterations are 1000. Also, the output is a Pareto front graph. The MOPSA works by defining a mesh grid around an initial point. This grid expands or shrinks depending on the previous search fitness. Figure 12 shows the Pareto front of both materials plus the selected optimal solutions. As mentioned before, the graph is a trade-off between the two objective functions; hence, one cannot say whether the MOPSA outperformed the MOGA as the MOPSA's solution is another feasible solution in the feasible area. Both materials have their optimal solution at high voltage and $f$ = 120 mm/min. However, the other three parameters are different; AISI 304 is optimal at $P_{off}$ = 6.44 μs, $P_{on}$ = 25 μs and $C$ = 1 A, while AISI 316 is optimal at $P_{off}$ = 6. μs, $P_{on}$ = 39.53 μs and $C$ = 3.34 A. For AISI 304, the *MRR* is 5.933 mm$^3$/min (4% higher than the MOGA's solution) and the surface roughness Ra is 4.448 μm (2% difference to the MOGA). In addition, the optimal *MRR* of AISI 316 is 5.96 mm$^3$/min and the optimal Ra is 4.676 μm (better surface quality and lower productivity than the MOGA's solution).

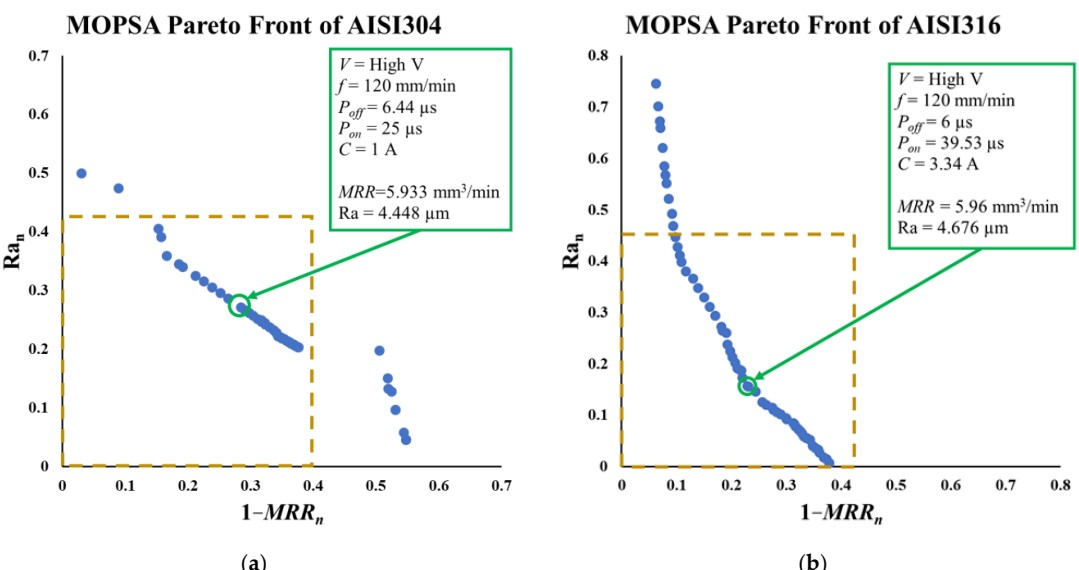

**Figure 12.** Optimal solutions by MOPSA Pareto front of both (**a**) AISI 304 and (**b**) AISI 316.

### 3.3.3. WVGWO Model

This algorithm is a bio-inspired simulation of the hunting journey of a wolf pack. Usually, the pack has an alpha wolf (leader); lower rank wolves, beta and delta; and the rest of the pack are called omega wolves. The WVGWO begins with the initial wolf pack (population) that is searching for prey (best solution) and moving together in a certain direction. If the pack find prey in a certain location (parameter set in the current population), this promotes the winner wolf to a higher rank in the next search. These promotions are stored in an archive during the whole searching process. This archive is named the non-dominated wolves, which are the red plotted data on Figure 13. The final population is called the grey wolves, which includes the non-dominated wolves. Similar to the previously illustrated algorithms, the non-dominated wolves are considered as the Pareto front date; however, the graph keeps the other solutions of the final population as grey wolves who cannot outperform the non-dominated wolves. Clearly, it is found that the optimal solution of the WVGWO is very similar to the MOPSA's solution (see Figures 12 and 13). Another remark, on achieving the same *MRR* for both stainless steel materials, AISI 304 attains better surface quality (around 4 to 5%) than AISI 316. Also, this final remark is found in the MOPSA model.

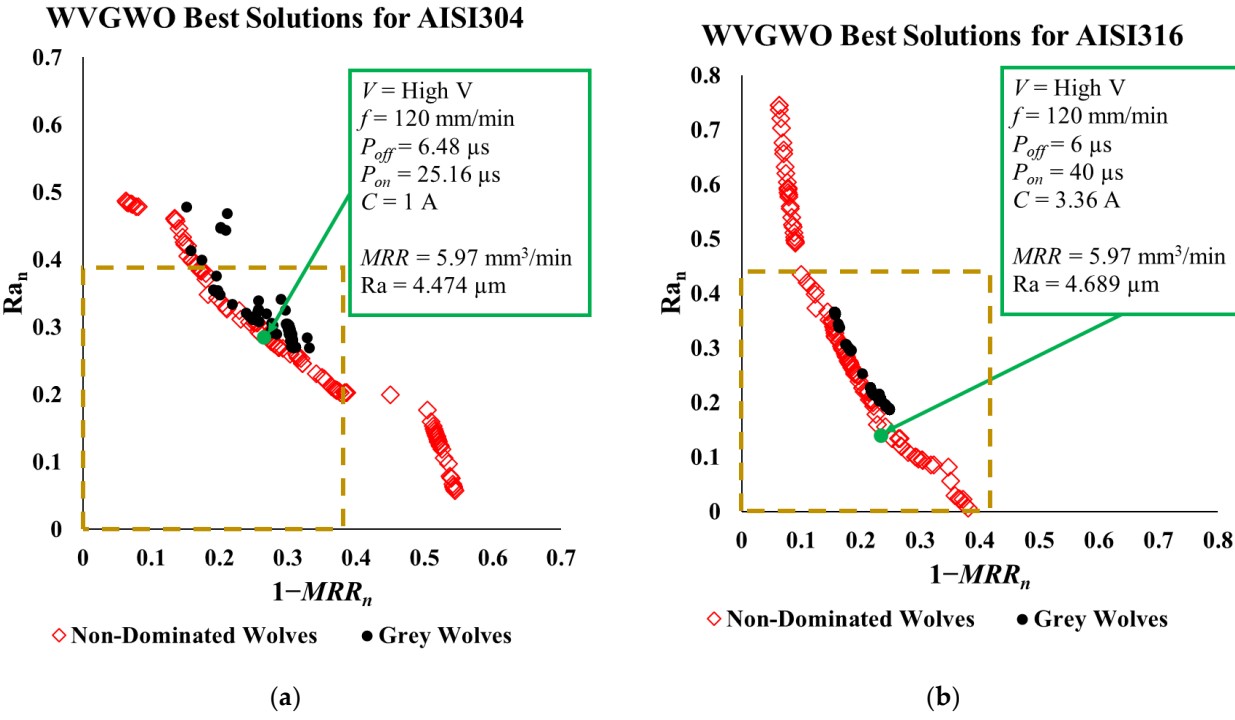

(**a**)  (**b**)

**Figure 13.** Optimal solutions by WVGWO of both (**a**) AISI 304 and (**b**) AISI 316.

### 3.3.4. OOA Model

Another bio-inspired algorithm is used called the osprey optimization algorithm (OOA). This algorithm is quite similar to the WVGWO; however, the hunting population moves separately and randomly, unlike the grey wolves that move together in one random direction. Ospreys start hunting fishes from underwater. The winner osprey indicates a hunt location that updates the current population by the evaluation of the objective function. The proposed MATLAB code by [42], obtained from the MathWorks website, creates one generation and finds the optimal solution after a certain number of iterations. In this work, a loop of generations is added while the optimal solution of each generation is stored in an archive. The optimal solutions are compared with the previous algorithms in a way that considers Ra as the first criterion. If a better surface quality solution is selected, the corresponding MRR solution is chosen—that is, the surface roughness objective of the OOA is the best amongst the algorithms studied; however, the MRR solution is lower than

that of the other algorithms. Table 6 provides the optimal solutions obtained by the OOA for both materials. Once more, the ratio between the surface quality of both materials is 4 to 5% (AISI 304 better than AISI 316) on achieving the same productivity (MRR).

**Table 6.** Optimal solutions by OOA of both stainless steels 304 and 316.

| Parameter | AISI 304 | AISI 316 |
|---|---|---|
| Voltage ($V$), V | High | High |
| Transverse feed ($f$), mm/min | 120 | 120 |
| Pulse-off time ($P_{off}$), μs | 6.38 | 6 |
| Pulse-on time ($P_{on}$), μs | 26.85 | 39.97 |
| Current intensity ($C$), A | 1 | 3.16 |
| Material removal rate ($MRR$), mm$^3$/min | 5.886 | 5.89 |
| Surface roughness (Ra), μm | 4.427 | 4.61 |

Table 6 entails the optimal solutions obtained by the OOA for both materials. Once more, the ratio between the surface quality of both materials is 4 to 5% (AISI 304 better than AISI 316) for achieving the same productivity (*MRR*).

To summarize, Scott et al. [45] concluded that no single combination can be an optimal solution for multi-responses of WEDM. However, the trade-off optimal solutions of both addressed responses (*MRR* and Ra) of all algorithms for both stainless steel materials are presented in Table 7.

**Table 7.** Optimal solutions comparison by all used algorithms.

| Model | AISI 304 | | | | | | | AISI 316 | | | | | | |
|---|---|---|---|---|---|---|---|---|---|---|---|---|---|---|
| | $V$ | $f$ | $P_{off}$ | $P_{on}$ | $C$ | $MRR$ | Ra | $V$ | $f$ | $P_{off}$ | $P_{on}$ | $C$ | $MRR$ | Ra |
| MOGA | High | 119.37 | 6.16 | 28.66 | 1.06 | 5.707 | 4.353 | High | 119.97 | 6.06 | 39.53 | 3.36 | 6 | 4.741 |
| MOPSA | High | 120 | 6.44 | 25 | 1 | 5.933 | 4.448 | High | 120 | 6 | 39.53 | 3.34 | 5.96 | 4.677 |
| WVGWO | High | 120 | 6.48 | 25.16 | 1 | 5.97 | 4.474 | High | 120 | 6 | 40 | 3.36 | 5.97 | 4.689 |
| OOA | High | 120 | 6.38 | 26.85 | 1 | 5.886 | 4.427 | High | 120 | 6 | 39.97 | 3.16 | 5.89 | 4.61 |

Obviously, both materials' optimal running voltage is high and feed rate is 120 mm/min. For AISI 304, mid-range pulse-off, low pulse-on and low current are optimal running parameters. On the other hand, the running conditions of AISI 316 are optimal at low pulse-off, high pulse-on and high current.

## 4. Conclusions

The findings of this optimization experimental-based study on the surface roughness and material removal rate of stainless steels 304 and 316 by WEDM are presented in this paper. The wire used in this research is molybdenum wire with a diameter of 0.18 mm. The investigated parameters are voltage, transverse feed rate, pulse-on/pulse-off times and current intensity. The surface roughness and material removal rate were evaluated and analyzed. Mathematical regression models were developed using MATLAB. Multi-objective optimization is carried out on these regression models. The main outputs of this research are as follows:

- Despite the fact that the WEDM process has a fuzzy proportion with the running parameters, the developed mathematical regression models represented the experimental results with small negligible errors that promote the models for optimization.
- The most influential parameters on both *MRR* and Ra are pulse-on time ($P_{on}$ and current ($C$).

- For the optimization model of AISI 304, the MOGA algorithm attained the best surface roughness at 4.353 μm, while the optimal *MRR* was obtained by the WVGWO at a value of 5.97 mm$^3$/min. However, the MOPSA provided a trade-off multi-response solution as Ra = 4.448 μm ($-2.18\%$ from the optimal solution by the MOGA) and *MRR* = 5.933 mm$^3$/min ($-0.62\%$ from the best solution by the WVGWO). The optimal parameters obtained by the MOPSA are high voltage, $f$ = 120 mm/min, $P_{off}$ = 6.44 μs, $P_{on}$ = 25 μs and $C$= 1 A.
- Similarly, for the AISI 316 model, the optimal Ra of 4.61 μm is obtained by the OOA and the optimal *MRR* = 5.97 mm$^3$/min by the WVGWO. Again, the MOPSA outperformed the other algorithms and resulted optimal *MRR* and Ra values of 5.96 mm$^3$/min and 4.677 μm, respectively. In this case, the obtained optimal parameters by the MOPSA are high voltage, $f$ = 120 mm/min, $P_{off}$ = 6 μs, $P_{on}$ = 39.53 μs and $C$ = 3.43 A.
- The optimal solution by the WVGWO of both materials in Table 7 show that the machining of AISI 304 and AISI 316 have the same productivity of *MRR* = 5.97 mm$^3$/min; however, AISI 304 has better surface roughness (Ra = 4.474 μm) than AISI 316 (Ra = 4.689 μm), making AISI 304 better by 4.58%.
- Obviously, the workpiece material's thermo-physical properties play a great role in the influence of WEDM parameters on the responses in terms of *MRR* and Ra.

**Author Contributions:** Conceptualization, A.E; methodology, M.A.A. and N.N.; formal analysis and investigation, A.E., M.A.A. and N.N.; writing—original draft preparation, I.H.A. and M.F.A.; writing—review and editing, I.H.A., M.F.A. and A.E.; optimization and modeling analysis, I.H.A. and M.F.A.; resources, A.E., M.A.A. and N.N.; supervision, A.E. and M.F.A. All authors have read and agreed to the published version of the manuscript.

**Funding:** This research received no external funding.

**Data Availability Statement:** The data presented in this study are available on request from the corresponding author.

**Conflicts of Interest:** The authors declare no conflict of interest.

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
