# Peer review of "Wire Electrical Discharge Machining of AISI304 and AISI316 Alloys: A Comparative Assessment of Machining Responses, Empirical Modeling and Multi-Objective Optimization"

_jmmp, doi:10.3390/jmmp7060194_

Round 1
Reviewer 1 Report
This research investigates multi-response of both material removal rate (MRR) and surface roughness (Ra) for the wire electrical discharge WEDM of two stainless steel alloys; AISI 304 and AISI 316. The experimental work is conducted through a full factorial design of experiment of five running parameters. Overall, the paper includes a small amount of newer contribution in the area of electrochemical machining of micro grooves.
I cannot agree with the conclusion in the article: ‘Eventually, the optimal results of both materials share the high voltage, the high transverse feed rate and low pulse-off time, however, AISI 304 requires a low level of pulse-on time and current intensity while AISI 316 optimal results entails a higher level of pulse-on time and current.’
For different magnitudes of experimental parameters, the experimental results will be biased. This article only chose to conduct research under specific limited parameters and cannot cover all laws. It is difficult to use reference data for industrialization.
The English grammar and punctuation of this manuscript should be more standardized and accurate.
Author Response
- This research investigates multi-response of both material removal rate (MRR) and surface roughness (Ra) for the wire electrical discharge WEDM of two stainless steel alloys; AISI 304 and AISI 316. The experimental work is conducted through a full factorial design of experiment of five running parameters. Overall, the paper includes a small amount of newer contribution in the area of electrochemical machining of micro grooves.
Answer: Thank you for this remark. We would like to reiterate the aim of this paper as follows;
“The aim of this study is to widely investigate and optimize the WEDM process parameters in order to achieve the trade-off between maximizing the MRR while keeping the Ra at the minimum level of two different steel alloys; stainless steel 304 and 316 (AISI 304 and AISI 316). A previously reported experimental investigation of WEDM of AISI 304 was experimentally extended for AISI 316 for which five parameters are investigated and modelled using MATLAB regression toolbox. A combination of four multi-objective algorithms is used and compared. The algorithms are; (1) multi-objective genetic algorithm (MOGA), (2) multi-objective pareto search algorithm (MOPSA), (3) weighted value grey wolf optimizer (WVGWO) and (4) osprey optimization algorithm”.
It is worth stating that the work carried out in this paper entails an extended experimental study to examine the effect of five process parameters on the machining responses of AISI 316. The results are compared with those obtained for the AISI 304. Furthermore, regression models were developed for both materials and multi-objective optimization techniques were utilized to fulfil the aforementioned aim in order to identify the working window of optimal process parameters that secure maximum MRR and minimum surface roughness. A number of insights were obtained and reported in the conclusion section which reflects the scientific contribution to the existing knowledge in the field.
- I cannot agree with the conclusion in the article: ‘Eventually, the optimal results of both materials share the high voltage, the high transverse feed rate and low pulse-off time, however, AISI 304 requires a low level of pulse-on time and current intensity while AISI 316 optimal results entails a higher level of pulse-on time and current.’
Answer: Thank you for your valuable comment. The optimal parameters for WEDM of these two materials, AISI 304 and AISI 316, differ in terms of pulse-on time and current intensity. While both materials require high voltage, high transverse feed rates, and low pulse-off times, the differences in pulse-on time and current intensity can be attributed to the distinct properties and compositions of these stainless-steel alloys.
- For different magnitudes of experimental parameters, the experimental results will be biased. This article only chose to conduct research under specific limited parameters and cannot cover all laws. It is difficult to use reference data for industrialization.
Answer: Thank you for this remark. We partially disagree with the reviewer comment. In particular, the range of process parameters and their levels were rationally selected based on previous experience, from industrial experts, with the working range of governing variables and the aim was to examine the effect of the range of these parameters on difference responses and thus to be able to identify even the most appropriate values to results in process optimum performance. These findings of the identified values still can be utilised for industrial implementation or in some sever industrial applications, same approached can be extended new values can be determined. However, it is worth stating that, we appreciate the reviewer concern and will extend the range of process parameters in our future work. Thank you once again for the critical and valuable remark.
Comments on the Quality of English Language
- The English grammar and punctuation of this manuscript should be more standardized and accurate.
Answer: Thank you for your comment. The entire manuscript was proofread, and all typos and punctuations were corrected.

Reviewer 2 Report
Dear Authors,
please consider the following aspects:
1. Line 54 – thermos?
2. Lines 59-62 - stated that the main benefits of using WEDM is good surface quality. In fact, other properties speak for the advantages of this method. After all, there are techniques to get much better surfaces. Here, the key should be to provide the possibility of processing materials that cannot be obtained by other techniques.
3. Line 101 – pulse off time? This seems to be a mistake. Pulse off time is the time interval between discharges. So does this affect the surface creation? Similarly in the next sentences (105, etc.)
4. Line 113 – what does it mean “optimal” – which values of MRR and roughness are expected?
5. Table 1 – units?
6. Table 3 - please add voltage values. What does it mean low or high voltage?
7. Table 5 – some values are missing.
Author Response
Reviewer 2
Dear Authors,
Please consider the following aspects:
- Line 54 – thermos?
Answer: Corrected.
- Lines 59-62 – stated that the main benefits of using WEDM is good surface quality. In fact, other properties speak for the advantages of this method. After all, there are techniques to get much better surfaces. Here, the key should be to provide the possibility of processing materials that cannot be obtained by other techniques.
Answer: The provided information is modified as follows;
“One of the advantages of WEDM is to process difficult-to-cut materials irrespective to their hardness with acceptable surface properties and accurate dimensions which promotes WEDM as a primal cutting method to produce punch dies and molds of high strength materials [1], [8], [10]”.
- Line 101 – pulse off time? This seems to be a mistake. Pulse off time is the time interval between discharges. So, does this affect the surface creation? Similarly in the next sentences (105, etc.)
Answer: The pulse off-time is the period of the rest or pause necessary for dielectric reionization. During this period, the molten material solidifies and is rinsed out of the spark gap. If the pulse off-time is too short, sparks become unstable, resulting in greater short-circuiting. Contrarily, a larger pulse off-time results in longer machining time, but it can provide the stability needed to effectively EDM a certain application. When the pulse off-time is insufficient in comparison to the on-time, unpredictable cycling and retraction of the advancing servo motor occur, slowing down the operation.
- Line 113 – what does it mean “optimal” – which values of MRR and roughness are expected?
Answer: This statement regarding the aim of this research is elaborated as follows;
“The aim of this study is to widely investigate and optimize the WEDM process parameters in order to achieve the trade-off between maximizing the MRR while keeping the Ra at the minimum level of two different steel alloys; stainless steel 304 and 316 (AISI 304 and AISI 316)”.
- Table 1 – units?
Answer: The numbers are percentages of elements exist in the alloys used. The percentage mark is added to the table caption.
- Table 3 - please add voltage values. What does it mean low or high voltage?
Answer: Thank you for your valuable comment. For the high and low levels of the voltage, as you can see in the following picture of the machine control software, the voltage can be varied between high and low, and looking in the catalogue it has no explanation but searching further in the literature for quite similar machines, but not the exact one, we found that, more likely high is equivalent to 200 V while low is for 80 V. We do not want to add these values in the manuscript as we not 100% sure of these values. I hope the reviewer will understand and accept our justification for this comment.
Machine readings image is attached in the doc. file.
- Table 5 – some values are missing.
Answer: In fact, the linear inequality and linear equality are represented by blank brackets when they are zeros.

Reviewer 3 Report
Brief summary
The aim of the paper is to investigate the multi-response of both material removal rate (MRR) and surface 20 roughness (Ra) for the wire electrical discharge WEDM of two stainless steel alloys. The main contribution of the paper is a comparison of different optimisation algorithms for analysing process data. The extensive experiment tests, as an outcome of DoE, are definitely a strength of this paper.
General concept comments
The paper is well structured and has a logical order. All references are up-to-date and relevant to the topic discussed in the paper. The research hypothesis is however not clearly stated. There is no clear information as to why these two materials were chosen for cutting from among many that are cut using WEDM worldwide. This should be better explained.
The question is why the authors used only a Ra parameter for surface roughness evaluation. This parameter is very insensitive to surface outliers. It would be better to use Rsk and/or Rku, as well as functional parameters, e.g., Rpk, Rk, Rvk. Further, it would be even better to measure and evaluate the surface in 3D.
The question is why the Ra parameter is so high? In modern WEDM machining, the surface roughness is achievable below 1 micrometre. Could authors explain the reason for so high surface roughness?
The visualisation of the measurements and correlations looks good, however, it is really hard to understand some of them.
Figure 2 should be presented as a table, as for now is way too big for the information that needs to be provided.
Figures 3 to 6 are a bit confusing. It is not clear how the heatmap image is created, or to be specific how many experiments were done to build it. For example, in Figure 4, the Ra parameter is fluctuating. However, Ra is normally just one value, so to build this Figure it requires a lot of single measurement data from different parameters. This needs to be clarified in the paper.
The optimisation algorithms described in the paper are giving similar results in optimised process parameters despite the processed material. This might indicate that the trial requires a more in-depth study. I strongly recommend using different materials to be compared to the chosen two.
How optimised parameters were validated?
The last thing is that the authors did not provide any information about measurement uncertainty, both for surface roughness, and MRR, as well as for computational models. This should be explained/mentioned in the paper.
Table 7 should have a vertical line separating the two materials.
Author Response
Reviewer 3
Brief summary
The aim of the paper is to investigate the multi-response of both material removal rate (MRR) and surface 20 roughness (Ra) for the wire electrical discharge WEDM of two stainless steel alloys. The main contribution of the paper is a comparison of different optimization algorithms for analyzing process data. The extensive experiment tests, as an outcome of DoE, are definitely a strength of this paper.m
General concept comments
- The paper is well structured and has a logical order. All references are up-to-date and relevant to the topic discussed in the paper. The research hypothesis is however not clearly stated. There is no clear information as to why these two materials were chosen for cutting from among many that are cut using WEDM worldwide. This should be better explained.
Answer: Thank you for this remarkable comment. This explanation should have been provided in the manuscript. The following text is added in the materials section 2.1.
Stainless steel is the most commonly utilized material for contact surfaces in dairy processing equipment. This metal possesses corrosion resistance, mechanical strength, hardness, and ease of manufacture (weldability). AISI type 304 and 316 are the most suit-able grades for general process fluid heating, storage, and distribution. Because of the presence of molybdenum, type 316 is more expensive but offers superior corrosion resistance. Given that it is primarily required to protect the machine from the atmosphere, water, and any spilled liquids, AISI 304 is almost always utilized externally, or for the outside vessel jacket. Not only are they known for their resistance to corrosion, they are also known for their clean appearance and overall cleanliness. [32].
[32] K. Cronin and R. Cocker, “Plant and Equipment | Materials and Finishes for Plant and Equipment,” in Encyclopedia of Dairy Sciences, Elsevier, 2011, pp. 134–138. doi: 10.1016/B978-0-12-374407-4.00401-5.
- The question is why the authors used only a Ra parameter for surface roughness evaluation. This parameter is very insensitive to surface outliers. It would be better to use Rsk and/or Rku, as well as functional parameters, e.g., Rpk, Rk, Rvk. Further, it would be even better to measure and evaluate the surface in 3D.
Answer: Thank you for your valuable comment. The authors' decision to use only the Ra (Arithmetic Average Roughness) parameter for surface roughness evaluation in our study might be influenced by several factors, and it's important to consider the context and purpose of their research when assessing this choice. Here are some potential reasons for their decision:
- Simplicity and Common Usage: Ra is one of the most commonly used surface roughness parameters in industry and research. It provides a simple and standardized way to describe surface roughness, making it easier for others to understand and compare their results. It serves as a baseline parameter that many people are familiar with.
- Application-Specific: The choice of roughness parameter can depend on the specific application or industry. Some applications may require more detailed information about the surface roughness, while others may find Ra sufficient. If the authors' research was focused on a particular industry or application where Ra is the standard, they may have chosen it for compatibility and ease of comparison.
The choice of surface roughness parameters depends on the specific goals, constraints, and context of the research. While 3D measurements and more advanced parameters like Rsk, Rku, Rpk, Rk, and Rvk can provide additional insights into surface roughness, Ra remains a widely accepted and practical parameter for many applications. Researchers often need to strike a balance between the depth of analysis and practical considerations when selecting roughness parameters for their work.
- The question is why the Ra parameter is so high? In modern WEDM machining, the surface roughness is achievable below 1 micrometer. Could authors explain the reason for so high surface roughness?
Answer: Thank you very much for the very critical question. Of course, we can reach such values, if we carefully had a strong influence on surface and through lead wire EDM with critical pulse durations of 20 to 200 µs. Nevertheless, first of all it must be clarified what kind of material to be cut, and wether surface roughness is the only parameters to be looked at and thus to scarify other responses such as MRR, which was not the case here. In particular, in our application we tried to find a compromise set of process parameters that enable high MRR while keeping the surface roughness at acceptable level. Our plan for future work entails a rigorous selection of dominant process parameters that can lead to either maximum level MRR or minimum level of surface roughness for which Ra below 1 micron will be targeted and expected.
- The visualization of the measurements and correlations looks good; however, it is really hard to understand some of them.
Answer: Figs 7-10 show the comparison between experimental results as blue circle marks and the mathematical regression model as an orange spline curve. The statistical analysis is provided for all figures indicating the R-squared, R-adjusted, p-value and the maximum relative error observed between the experiment and model results. For MRR, the correlation appears to have a good fit while the Ra correlation has acceptable fit as it has significant p-values.
- Figure 2 should be presented as a table, as for now is way too big for the information that needs to be provided.
Answer: This figure can be reduced in size; however, the authors find it important to show the full factorial design of experiment with different factors levels 23×32.
- Figures 3 to 6 are a bit confusing. It is not clear how the heatmap image is created, or to be specific how many experiments were done to build it. For example, in Figure 4, the Ra parameter is fluctuating. However, Ra is normally just one value, so to build this Figure it requires a lot of single measurement data from different parameters. This needs to be clarified in the paper.
Answer: These figures are representations of the effect of five process parameters on the MRR and Ra. Hence, in order to reduce the number of dimensions to two “pulse-on time and current”, the results are divided into 8 subfigures for each material response. Each figure appears like a hierarchy representation. The first two columns indicate the voltage and feed rate, while the 3rd and 4th columns indicate the pulse-off time. For example, Figure 4e shows the effect of low voltage, feed rate of 80 mm/min and pulse-off time of 6 µs on the Ra of AISI 304 alloy with variations of the pulse-one time on x-axis and current on y-axis.
- The optimisation algorithms described in the paper are giving similar results in optimised process parameters despite the processed material. This might indicate that the trial requires a more in-depth study. I strongly recommend using different materials to be compared to the chosen two.
Answer: In fact, the optimization results exhibit similarity in the voltage and feed rate as these parameters affect the WEDM process generally no matter what the processed material is. However, the other 3 parameters are totally different for each material. It is agreed that all algorithms searched similar optima, however, the optima are different for the AISI 304 and AISI 316 as shown in Table 7.
- How optimised parameters were validated?
Answer: The optimal results are localized in the range of the experimental results as shown in the heatmap figures (Figs 3 to 6). Also, the optimal results are matching the experimental output. Hence, it was unnecessary to carry out the validation trials.
- The last thing is that the authors did not provide any information about measurement uncertainty, both for surface roughness, and MRR, as well as for computational models. This should be explained/mentioned in the paper.
Answer: Thank you for this comment. The “Measurement and characterization” Section was modified as follows to briefly discuss the measurement uncertainty.
“Each experimental trial of certain set of cutting parameters had two wire cuts. The first cut is 15 mm length that is machined in order to evaluate the cut width. The machined specimens were photographed by a high-resolution camera under appropriate lighting conditions. Following that, the collected images are processed and analyzed using the image processing techniques as presented in [10] to detect the edge and precise width of the cut slot along its entire length. In addition, the machining time is calculated alongside the cut width. This allowed the material removal rate (MRR) to be calculated. The second cut is a full-width cut of 30 mm that allowed the characterization of the surface quality of the side walls of the cut. A Mitutoyo SJ-210 surface profilometer (Mitutoyo Corp., Kawasaki, Japan) is used to measure the surface roughness (Ra) along the full length of the side walls. To eliminate measurement uncertainty, seven readings are measured for each specimen, and the average measurement is considered, further details of the characterization procedures can be found in [10].”
While the accuracy of the created regression models was discussed as part of the results in Sections 3.2.1 and 3.2.2.
- Table 7 should have a vertical line separating the two materials.
Answer: It is modified.

Round 2
Reviewer 1 Report
no